# Most bothersome symptom in migraine and probable migraine: A population-based study

**Seung Jae Kim**[1◉], **Hye Jeong Lee**[2,3◉], **Sue Hyun Lee**[4], **Soomi Cho**[1], **Kyung Min Kim**[1], **Min Kyung Chu**[1]*

1 Department of Neurology, Severance Hospital, Yonsei University College of Medicine, Seoul, Korea,
2 Department of Neurology, Yonsei University College of Medicine, Seoul, Korea, 3 Department of
Neurology, Chung-Ang University Gwangmyeong Hospital, Gwangmyeong, Gyeonggi-do, Korea,
4 Department of Neurology, Wonju Severance Christian Hospital, Yonsei University Wonju College of
Medicine, Wonju, Gangwon-do, Korea

◉ These authors contributed equally to this work.
* chumk@yonsei.ac.kr

doi.org/10.1371/journal.pone.0289729

Dokkyo Ika Daigaku, JAPAN

**Data Availability Statement:** The datasets
generated and analyzed during the current study
are available at https://doi.org/10.6084/m9.
figshare.23599749.

## Abstract

Recently, the most bothersome symptom has been recommended as a co-primary endpoint
in clinical trials on the acute treatment of migraine. Probable migraine is a subtype of
migraine that fulfills all but one criterion for migraine diagnosis. We aimed to compare the
most bothersome symptom between probable migraine and migraine. This study analyzed
data from a nationwide study conducted in Korea, and the most bothersome symptom was
assessed by requesting the participants to select one of the four typical accompanying
symptoms of migraine. Responses to acute treatment were evaluated using the migraine
Treatment Optimization Questionnaire-6. Nausea was the most bothersome symptom, fol-
lowed by phonophobia and vomiting in the migraine group (nausea, 61.8%; phonophobia,
25.3%; vomiting, 10.0%; and photophobia, 2.9%) and the probable migraine group (nausea,
82.2%; phonophobia, 9.5%; vomiting, 5.6%; and photophobia, 2.7%). In participants with
migraine, vomiting (adjusted odds ratio = 6.513; 95% confidence interval, 1.763–24.057)
and phonophobia (adjusted odds ratio = 0.437; 95% confidence interval, 0.206–0.929) were
significantly associated with severe headache intensity and nausea was significantly associ-
ated with >3 headache days per 30 days (adjusted odds ratio = 0.441; 95% confidence,
0.210–0.927). Different patterns of associations were observed in probable migraine.

## Introduction

Migraine is a complex of various symptoms in addition to headache; these include nausea,
vomiting, photophobia, and phonophobia [1]. Based on the current criteria, patients should
present with typical accompanying symptoms, such as nausea and/or vomiting or both photo-
phobia and phonophobia, to be diagnosed with migraine without aura (MO) [2]. Thus, the
reported migraine symptoms differ from person to person.

Apart from the suffering caused by headaches, various other accompanying symptoms of
migraine can cause additional difficulties. Thus, the United States Food and Drug

**Funding:** This research was supported by a grant from the Korea Health Technology R&D Project through the Korea Health Industry Development Institute (KHIDI), funded by the Ministry of Health & Welfare, Republic of Korea (Grant No.: HV22C0106) and a National Research Foundation of Korea (NRF) grant from the Korean government (MSIT) (2022R1A2C1091767). The funders had no role in study design, data collection and analysis, decision to publish, or preparation of the manuscript.

**Competing interests:** M.K.C. was a site investigator for a multicenter trial sponsored by Biohaven Pharmaceuticals, Allergan Korea, and Ildong Pharmaceutical Company. He received lecture honoraria from Eli Lilly and Company, Handok-Teva, and Ildong Pharmaceutical Company over the past 24 months. He received grants from Yonsei University College of Medicine (6-2021-0229), the Korea Health Industry Development Institute (KHIDI) (Grant No.: HV22C0106), and a National Research Foundation of Korea (NRF) grant from the Korean government (MSIT) (2022R1A2C1091767). No other authors possess competing interests. This does not alter our adherence to PLOS ONE policies on sharing data and materials.

Administration and the International Headache Society recommend the most bothersome symptom (MBS) as an efficacy endpoint for the acute treatment of migraine [3, 4]. The MBS has been used both as a co-primary endpoint to better align the study outcome with the symptoms that are of primary importance to the patients and to demonstrate an effect on both pain and the MBS [3]. In acute treatment clinical trials, use of the MBS can reflect "real-world" treatment outcomes more appropriately as compared to evaluating changes in pain alone [5, 6]. The MBS can provide insights into the relationship between the accompanying symptoms and the burden of migraine. Studies have revealed that acute treatment optimization, migraine symptom severity, and visual aura (VA) differ significantly according to the MBS [1, 7].

Probable migraine (PM) is a subtype of migraine that fulfills all but one of the criteria for migraine diagnosis. Epidemiological studies have revealed the presence of PM in a significant proportion of the general population [8, 9]. The symptoms and disability in PM differ from those in migraine; therefore, we hypothesized that the MBS of PM may differ from that of migraine. However, limited data are available on the MBS in individuals with PM. Therefore, we aimed to evaluate the MBS of participants with PM and compare it with the MBS of participants with migraine using data from a nation-wide population study.

## Materials and methods

### Study design and participants

We used baseline data from the Circannual Change in Headache and Sleep (CHASE) study, which was conducted in October 2020 [8]. The CHASE study is a nationwide population-based survey on circannual changes in headache and sleep in the Korean population. The target participants were adults aged 20–59 years. We used a two-stage clustered random sampling method (based on the Korean national census data from 2015) to obtain a sample proportional to the population distribution in all Korean territories [10]. To conduct a web-based survey, e-mails were sent to 91,153 individuals from the sample population; 3,030 responders were finally enrolled in this study.

The participants' sociodemographic data, headache characteristics, and sleep status were investigated during the baseline assessment phase of the CHASE study. We used questionnaires to evaluate the characteristics of headaches; disabilities secondary to headaches; and status of anxiety, depression, and stress. The data from the CHASE study do not contain personal information and cannot be used to identify the participants.

### Diagnosis of migraine and PM

Migraine and PM were diagnosed using a web-based questionnaire based on the International Classification of Headache Disorders, Third Edition (ICHD-3) [2, 11]. A migraine diagnostic module was constructed according to the ICHD-3 criteria for MO (code 1.1) as follows: (A) occurrence of at least five attacks fulfilling criteria B–D; (B) headache attacks lasting for 4–72 h (when untreated or treated unsuccessfully); (C) headaches with at least two of the following four characteristics: (1) unilateral location, (2) pulsating quality, (3) moderate-to-severe pain intensity, and (4) aggravated by or causing the avoidance of routine physical activity (e.g., walking or climbing stairs); (D) presence of at least one of the following symptoms during headache: (1) nausea and/or vomiting and (2) photophobia and phonophobia; and (E) attacks not better accounted for by another ICHD-3-based diagnosis. Furthermore, PM was diagnosed according to the following ICHD-3-based criterion (code 1.5): headaches fulfilling all but one of the A–E criteria for MO.

Migraine with aura (code 1.2) was diagnosed when headaches fulfilled the diagnostic criteria for both migraine with aura and MO. Therefore, in the present study, "migraine" refers to

both migraine with aura and MO. Similarly, PM refers to both PM with aura (code 1.5.2) and without aura (code 1.5.1). Our headache diagnostic module had an estimated sensitivity and specificity of 92.6% and 94.8% for the diagnosis of migraine, respectively; the corresponding values for the diagnosis of PM were 85.0% and 92.9%, respectively [11].

### Diagnosis of VA

VA was diagnosed using the validated, self-reporting Visual Aura Rating Scale (VARS) questionnaire [12]. The questionnaire is composed of five yes/no questions on visual symptoms; the response to each is rated on a scale of 1–3 points as follows: (1) duration of a symptom ranging from 5 min to 60 min (3 points); (2) symptom develops gradually ≥5 min (2 points); (3) scotoma (2 points); (4) zig-zag line (2 points); and (5) unilateral (1 point). VA was diagnosed in case of a total VARS score of ≥3. The sensitivity and specificity of the self-reporting VARS questionnaire (with reference to the ICHD-3-based physician diagnosis) were 96.4% and 79.5%, respectively [13].

### Assessment of the MBS

The participants were asked the following question to assess their MBS: "In the past year, which of the following was the most bothersome symptom that accompanied your headache?" They could select only one of the following four symptoms in response: nausea, vomiting, photophobia, and phonophobia.

### Assessment of headache-related disability

Headache-related disability was evaluated using the Migraine Disability Assessment (MIDAS) questionnaire [14]. This five-item instrument evaluates the impact of migraine on the respondent's daily activities over the past 3 months. The MIDAS score is calculated by adding the number of days the headache interfered with work, school, household chores, or social activities. The Korean version of the MIDAS questionnaire has been previously validated [15].

### Assessment of depression, anxiety, and stress symptoms

Depressive symptoms were evaluated using the Patient Health Questionnaire-9 (PHQ-9) [16]. The PHQ-9 comprises nine questions for evaluating the severity of the respondent's depressive symptoms over the past 2 weeks. Depression was diagnosed in the case of PHQ-9 scores of ≥10. Anxiety symptoms were evaluated using the Generalized Anxiety Disorder-7 (GAD-7) scale [17]. The GAD-7 is a seven-item questionnaire designed to assess the severity of the respondent's anxiety. Anxiety was diagnosed in the case of GAD-7 scores of ≥8. The Korean versions of the PHQ-9 and GAD-7 have been previously validated [18, 19].

### Assessment of the response to acute treatment for migraine and PM

We assessed the participants' responses to acute treatment upon receiving a positive reply to the question "Do you use medications to treat headache when headache occurs?". Acute treatment response was measured using the migraine Treatment Optimization Questionnaire (mTOQ-6) [20]. This is a self-administered questionnaire on the frequency of favorable outcomes achieved by acute treatment. It comprises six questions on the following items: rapid return to function, 2 h pain-free, 24 h sustained pain relief, tolerability, ability to make plans, and perceived control. Each item was scored as follows: never (1), rarely (2), less than half the time (3), and more than half the time (4); accordingly, the total scores ranged from 1 to 24.

## Statistical analysis

Categorical variables are expressed as frequencies and percentages; these were analyzed using the chi-square test or Fisher's exact test. The normality of continuous data was assessed using the Kolmogorov–Smirnov normality test. Normally distributed continuous variables are expressed as means and standard deviations; these were analyzed using the Student's $t$-test or an analysis of variance. Non-normally distributed continuous variables are expressed as medians and interquartile ranges; these were analyzed using the Mann–Whitney $U$ test or the Kruskal–Wallis test, as appropriate. To compare the age- and sex-adjusted demographic and clinical characteristics between each MBS, the categorical and continuous variables were subjected to a logistic regression analysis and an analysis of covariance (ANCOVA), respectively. We performed a parametric ANCOVA if the model did not satisfy the parametric assumptions because it was robust when the assumptions were violated [21].

Multiple logistic regression analyses were conducted to investigate the correlations between the demographic and clinical characteristics of each MBS. In these regression analyses, we assessed age ($\geq$40 vs <40 years), headache intensity (mild-to-moderate vs. severe), and the MIDAS score ($\geq$11 vs <11) as dichotomous variables. Firth's logistic regression was performed upon detecting zero observations in a group [22]. The results are expressed as adjusted odds ratios (AORs) with 95% confidence intervals (CIs). Since our web-based survey required responses to all items, there were no missing data. All statistical analyses were conducted using the Statistical Package for Social Sciences (version 25.0; IBM, Armonk, NY, USA) and R (version 3.5.3) [23]. A two-sided $p$-value of <0.05 was considered statistically significant.

## Ethics declarations

The present study is a secondary analysis of anonymized data from a previous study (i.e., the CHASE study), which was approved by the institutional review board of the Severance Hospital, Yonsei University (approval no.: 2020-0038-001). All participants provided written informed consent before participation. This study was conducted in accordance with the tenets of the Declaration of Helsinki and relevant regulations.

## Results

### Participants

Overall, 3,030 participants were enrolled in this study (participation rate: 28.3% [3,030/10,699]); among these, 1,938 reported experiencing at least one headache episode in the previous year. Furthermore, 170 (5.6%) and 339 (11.2%) participants were diagnosed with migraine and PM, respectively. Among those with PM, 337 and 2 were diagnosed based on missing typical duration and missing typical accompanying symptoms, respectively. Fig 1 summarizes the flow of participants. The sex, age, size of residential area, and education level did not differ significantly between the participants and the general population of Korea. Furthermore, participants with migraine and those with PM differed significantly in terms of sex, but not in terms of age, size of residential area, and education level (S1 Table).

### Demographic and clinical characteristics of participants with migraine and those with PM

The headache days per 30 days; severe headache days per 30 days; days on acute medication per 30 days; and the MIDAS, PHQ-9, and GAD-7 scores were significantly higher in participants with migraine than in those with PM. Meanwhile, the mTOQ-6 scores were significantly higher in participants with PM than in those with migraine (Table 1).

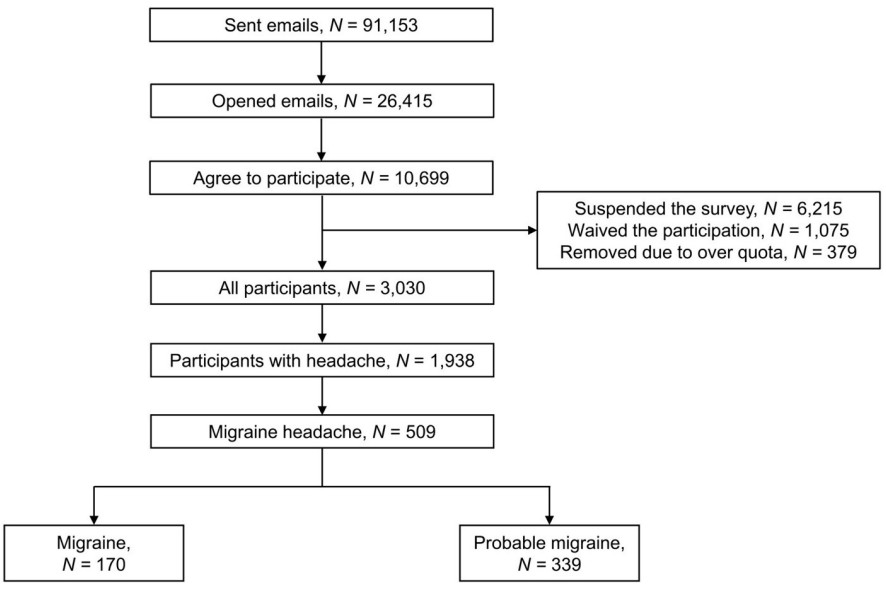

**Fig 1. A schematic of the participant flow.**

**Table 1. Demographic and clinical characteristics of participants with migraine and those with PM.**

| | Migraine, n = 170 | PM, n = 339 | *p*-value |
|---|---|---|---|
| **Demographic characteristics** | | | |
| Age (years) | 40.0 (32.8–46.0) | 41.0 (31.0–49.0) | 0.501 |
| Women | 129 (75.9) | 207 (61.1) | 0.001 |
| **Clinical characteristics** | | | |
| Headache days per 30 days | 3.0 (2.0–6.0) | 2.0 (1.0–4.0) | 0.006 |
| Severe headache days per 30 days | 2.0 (1.0–4.0) | 1.0 (1.0–3.0) | 0.004 |
| Days on acute medications per 30 days | 2.0 (1.0–5.0) | 1.0 (1.0–3.0) | <0.001 |
| Unilateral pain | 91 (53.5) | 241 (71.1) | <0.001 |
| Pulsating quality | 107 (62.9) | 249 (73.5) | 0.034 |
| Severe headache intensity | 79 (46.5) | 10 (3.0) | <0.001 |
| Aggravation by movement | 138 (81.2) | 184 (54.3) | <0.001 |
| Nausea | 120 (70.6) | 282 (83.2) | 0.001 |
| Vomiting | 82 (48.2) | 132 (38.9) | 0.022 |
| Photophobia | 125 (73.5) | 146 (43.1) | <0.001 |
| Phonophobia | 137 (80.6) | 166 (48.9) | <0.001 |
| MIDAS score | 13.0 (6.0–28.0) | 6.0 (2.0–15.0) | <0.001 |
| m-TOQ[a] score | 22.0 (17.0–25.0) | 23.0 (19.0–26.0) | 0.029 |
| Anxiety (GAD-7 score) | 6.0 (3.0–9.0) | 5.0 (2.0–8.0) | 0.016 |
| Depression (PHQ-9 score) | 8.0 (5.0–9.0) | 7.0 (4.0–9.0) | 0.035 |
| Visual aura | 50 (29.4) | 82 (24.1) | 0.219 |

Note: Data are expressed as n (%) or median (25%–75%)

PM, probable migraine; MIDAS, Migraine Disability Assessment; m-TOQ, Migraine Treatment Optimization Questionnaire; GAD-7, Generalized Anxiety Disorder-7; PHQ-9, Patient Health Questionnaire-9

[a]Only 127 and 213 participants responded to the m-TOQ for migraine and PM, respectively.

### Accompanying symptoms of migraine and PM

Among the 170 participants with migraine, phonophobia was the most frequently accompanying symptom (80.6% [137/170]), followed by photophobia (73.5% [125/170]), nausea (70.6% [120/170]), and vomiting (48.2% [82/170]). Participants reporting two of the four accompanying symptoms were predominant in the migraine group (47.1% [80/170]). Among the 339 participants with PM, nausea was the most frequently accompanying symptom (83.2% [282/339]), followed by phonophobia (49.0% [166/339]), photophobia (43.1% [146/339]), and vomiting (39.0% [132/339]). Participants reporting two of the four accompanying symptoms were predominant in the PM group (39.8% [135/339]). These data are summarized in S2 Table.

### Distribution of the MBS in the migraine and PM groups

Nausea was the most common MBS in both the migraine (61.8% [105/170]) and PM (81.7% [277/339]) groups. Phonophobia was the second most common MBS in both the migraine (25.3% [43/170]) and PM (10.0% [32/339]) groups. Photophobia was the least common MBS in both the migraine (2.9% [5/170]) and PM (2.7% [9/339]) groups. Multiple logistic regression analysis, adjusted for age and sex, revealed that vomiting (AOR = 0.484; 95% CI, 0.241–0.972; $p$ = 0.041) and phonophobia (AOR = 0.332; 95% CI, 0.200–0.522; $p < 0.001$) as the MBSs were less common in participants with PM than in those with migraine. In contrast, nausea (AOR = 2.768; 95% CI, 1.817–4.216; $p < 0.001$) as the MBS was more common in participants with PM than in those with migraine (Table 2).

### Associations of the MBS with the demographic and clinical characteristics in the migraine and PM groups

Multiple logistic regression analysis, adjusted for age and sex, revealed that vomiting as the MBS was positively associated with severe headache intensity (AOR = 6.513; 95% CI, 1.763–24.057, $p$ = 0.005) among participants with migraine. Conversely, phonophobia was negatively associated with severe headache intensity (AOR = 0.437; 95% CI, 1.763–24.057; $p$ = 0.031) and nausea was negatively associated with >3 headache days per month (AOR = 0.441; 95% CI, 0.210–0.927, $p$ = 0.031) in this group (Table 3). No significant associations between the MBS and the demographic and clinical characteristics were observed among participants with PM (Table 4).

### Response to acute treatment according to the MBS in the migraine and PM groups

Because the MBS was initially adopted in clinical trials on the acute treatment of migraine by the US Food and Drug Administration, we evaluated responses to the acute treatment of migraine and PM using the mTOQ-6 [3]. Among the 170 participants with migraine and the 337 participants with PM, 127 (74.7%) and 213 (63.2%) participants used medications for acute treatment, respectively (S3 Table). The responses to acute treatment did not differ significantly according to the MBS between the migraine and PM groups (Table 5).

## Discussion

Our primary findings were as follows: 1) nausea was the most common MBS, followed by phonophobia, vomiting, and photophobia, in both migraine and PM; 2) certain clinical characteristics were significantly associated with specific MBSs in migraine but not in PM; and 3) responses to the acute treatment of migraine and PM were not significantly associated with the MBS.

**Table 2. Comparison of the MBS between participants with migraine and those with PM and results of the logistic regression analysis for predicting PM for each MBS.**

| | Nausea | Vomiting | Photophobia | Phonophobia |
|---|---|---|---|---|
| Migraine, n (%) | 105 (61.8) | 17 (10.0) | 5 (2.9) | 43 (25.3) |
| PM, n (%) | 277 (81.7) | 19 (5.6) | 9 (2.7) | 34 (10.0) |
| PM[a]; OR (95% CI), p-value | 2.768 (1.817–4.216), <0.001 | 0.484 (0.241–0.972), 0.041 | 0.994 (0.324–3.048), 0.991 | 0.332 (0.200–0.552), <0.001 |

MBS, most bothersome symptom; PM, probable migraine; OR, odds ratio; CI, confidence interval

[a]Reference group

**Table 3. Demographic and clinical characteristics of participants with migraine according to the MBS.**

| | MBS in migraine | | | |
|---|---|---|---|---|
| | Nausea, n = 105 | Vomiting, n = 17 | Photophobia, n = 5[*] | Phonophobia, n = 43 |
| **Migraine subtype** | | | | |
| Migraine without VA[†], n (%) | 76 (63.3) | 14 (11.7) | 2 (1.7) | 28 (23.3) |
| Migraine with VA, n (%) | 29 (58.0) | 3 (6.0) | 3 (6.0) | 15 (30.0) |
| OR (95% CI) | 0.810 (0.394–1.665) | 0.451 (0.114–1.783) | 4.486 (0.809–28.861) | 1.332 (0.602–2.947) |
| p-value | 0.567 | 0.256 | 0.084 | 0.480 |
| **Age** | | | | |
| >40 years, n (%) | 45 (55.6) | 9 (11.1) | 4 (4.9) | 23 (28.4) |
| ≤40[†] years, n (%) | 60 (67.4) | 8 (9.0) | 1 (1.1) | 20 (22.5) |
| OR (95% CI) | 0.542 (0.277–1.057) | 1.392 (0.464–4.170) | 2.674 (0.440–28.332) | 1.579 (0.745–3.344) |
| p-value | 0.072 | 0.555 | 0.296 | 0.233 |
| **Sex** | | | | |
| Women, n (%) | 75 (58.1) | 13 (10.1) | 5 (3.9) | 36 (27.9) |
| Men[†], n (%) | 30 (73.2) | 4 (9.8) | 0 (0.0) | 7 (17.1) |
| OR (95% CI) | 0.523 (0.231–1.182) | 0.652 (0.178–2.387) | 3.988 (0.397–536.229) | 2.195 (0.842–5.722) |
| p-value | 0.119 | 0.518 | 0.281 | 0.108 |
| **Headache intensity** | | | | |
| Mild-to-moderate[†], n (%) | 58 (63.7) | 3 (3.3) | 1 (1.1) | 29 (31.9) |
| Severe, n (%) | 47 (59.5) | 14 (17.7) | 4 (5.1) | 14 (17.7) |
| OR (95% CI) | 0.842 (0.442–1.605) | 6.513 (1.763–24.057) | 3.259 (0.587–32.758) | 0.437 (0.206–0.929) |
| p-value | 0.601 | 0.005 | 0.184 | 0.031 |
| **Headache days** | | | | |
| >3.0, n (%) | 38 (53.5) | 7 (9.9) | 2 (2.8) | 24 (33.8) |
| ≤3.0[†], n(%) | 67 (67.7) | 10 (10.1) | 3 (3.0) | 19 (19.2) |
| OR (95% CI) | 0.441 (0.210–0.927) | 1.261 (0.371–4.288) | 1.035 (0.121–8.890) | 2.504 (1.091–5.747) |
| p-value | 0.031 | 0.710 | 0.974 | 0.030 |
| **MIDAS score** | | | | |
| >11, n (%) | 59 (60.2) | 10 (10.2) | 3 (3.1) | 26 (26.5) |
| <11[†], n (%) | 46 (63.9) | 7 (9.7) | 2 (2.8) | 17 (23.6) |
| OR (95% CI) | 1.225 (0.597–2.516) | 1.012 (0.309–3.311) | 0.931 (0.116–7.225) | 0.783 (0.344–1.781) |
| p-value | 0.580 | 0.985 | 0.943 | 0.559 |

Note: Data are expressed as n (%) or median (25%–75%)

MBS, most bothersome symptom; VA, visual aura; OR, odds ratio; CI, confidence interval; MIDAS, Migraine Disability Assessment

[*]No participant with severe headache intensity reported photophobia as the MBS. Firth's logistic regression was used because of the presence of cells with zero count.

[†]Reference group

**Table 4. Demographic and clinical characteristics of participants with PM according to the MBS.**

| | | MBS in PM | | | |
|---|---|---|---|---|---|
| | | Nausea, n = 277 | Vomiting, n = 19 | Photophobia, n = 9[*] | Phonophobia, n = 34 |
| **Migraine subtype** | | | | | |
| | Migraine without VA[†], n (%) | 213 (83.2) | 13 (5.1) | 5 (2.0) | 25 (9.8) |
| | Migraine with VA, n (%) | 64 (77.1) | 6 (7.2) | 4 (4.8) | 9 (10.8) |
| | OR (95% CI) | 0.647 (0.340–1.232) | 1.356 (0.460–3.992) | 2.618 (0.671–9.694) | 1.207 (0.506–2.879) |
| | p-value | 0.185 | 0.581 | 0.158 | 0.672 |
| **Age** | | | | | |
| | >40 years, n (%) | 143 (82.7) | 11 (6.4) | 3 (1.7) | 16 (9.2) |
| | ≤40 years, n (%) | 134 (80.7) | 8 (4.8) | 6 (3.6) | 18 (10.8) |
| | OR (95% CI) | 1.283 (0.722–2.279) | 1.386 (0.525–3.659) | 0.508 (0.119–1.844) | 0.660 (0.312–1.399) |
| | p-value | 0.395 | 0.510 | 0.306 | 0.279 |
| **Sex** | | | | | |
| | Women, n (%) | 170 (81.3) | 8 (3.8) | 6 (2.9) | 25 (12.0) |
| | Men[†], n (%) | 107 (82.3) | 11 (8.5) | 3 (2.3) | 9 (6.9) |
| | OR (95% CI) | 0.928 (0.518–1.664) | 0.452 (0.174–1.176) | 1.240 (0.347–5.250) | 1.790 (0.794–4.035) |
| | p-value | 0.803 | 0.104 | 0.746 | 0.160 |
| **Headache intensity** | | | | | |
| | Mild-to-moderate[†], n (%) | 271 (82.6) | 17 (5.2) | 9 (2.7) | 31 (9.5) |
| | Severe, n (%) | 6 (54.5) | 2 (18.1) | 0 (0.0) | 3 (27.3) |
| | OR (95% CI) | 0.255 (0.068–0.960) | 4.983 (0.885–28.057) | 2.563 (0.019–26.250) | 2.722 (0.527–14.059) |
| | p-value | 0.043 | 0.069 | 0.580 | 0.232 |
| **Headache days** | | | | | |
| | >3.0, n (%) | 74 (80.4) | 7 (7.6) | 4 (4.3) | 7 (7.6) |
| | ≤3.0[†], n (%) | 203 (82.9) | 12 (4.9) | 5 (2.0) | 27 (10.9) |
| | OR (95% CI) | 0.914 (0.457–1.827) | 1.548 (0.511–4.690) | 1.817 (0.396–7.936) | 0.692 (0.260–1.842) |
| | p-value | 0.799 | 0.440 | 0.432 | 0.461 |
| **MIDAS score** | | | | | |
| | >11, n (%) | 88 (80.0) | 8 (7.3) | 4 (3.6) | 10 (9.1) |
| | <11[†], n (%) | 189 (82.5) | 11 (4.8) | 5 (2.2) | 24 (10.5) |
| | OR (95% CI) | 0.880 (0.453–1.708) | 1.292 (0.436–3.829) | 1.025 (0.215–4.486) | 1.022 (0.422–2.477) |
| | p-value | 0.705 | 0.644 | 0.974 | 0.962 |

Note: Data are expressed as n (%) or median (25%–75%)

PM, probable migraine; MBS, most bothersome symptom; VA, visual aura; OR, odds ratio; CI, confidence interval; MIDAS, Migraine Disability Assessment

[*]No participant with severe headache intensity reported photophobia as the MBS. Firth's logistic regression was used because of the presence of cells with zero count.

[†]Reference group

**Table 5. Response to acute treatment according to the MBS in the migraine and PM groups.**

| | MBS | | | | |
|---|---|---|---|---|---|
| mTOQ-6 score | Nausea | Vomiting | Photophobia | Phonophobia | p-value[†] |
| Migraine | 22.0 (17.3–24.0) | 20.0 (14.8–24.0) | 22.0 (10.0–22.0) | 22.5 (17.0–26.3) | 0.257 |
| PM | 23.0 (19.0–27.0) | 22.0 (19.3–23.0) | 23.0 (15.8–25.2) | 21.0 (17.6–25.0) | 0.392 |

Note: Data are expressed as median (25%–75%)

MBS, most bothersome symptom; PM, probable migraine; mTOQ-6, Migraine Treatment Optimization Questionnaire-6

[†]Analysis of covariance was conducted after adjusting for age and sex

We requested participants to select among nausea, vomiting, phonophobia, and photophobia for assessing the MBS, because the ICHD-3 has defined these as the typical accompanying symptoms of migraine.

The current diagnostic criteria for migraine require the presentation of typical accompanying symptoms in addition to the presentation of the typical headache characteristics. Thus, individuals with migraine are reported to experience these accompanying symptoms along with headaches. Consequently, identification of the MBS among the accompanying symptoms may provide an understanding of the experiences of individuals with migraine. Therefore, we evaluated the MBS from among the four cardinal accompanying symptoms.

Previously, the MBSs have been evaluated in European countries and in the United States. Two large-scale American studies revealed photophobia as the most common MBS, followed by phonophobia; vomiting was the least common MBS [1, 24]. Furthermore, clinical trials investigating the MBS in European countries and America revealed that photophobia was the most common MBS, followed by nausea and phonophobia [25–28]. To date, only one Asian study has reported on the MBS in a population with migraine. A Taiwanese hospital study revealed that nausea was the most common MBS, followed by phonophobia and photophobia [7]. Our results are similar to those of this Taiwanese study, in that nausea was the most common MBS, followed by phonophobia. The reasons for underlying these regional differences in the MBS is are difficult to elucidate. One possible reason is the difference in the tolerance thresholds for each migraine symptom between the Asian and Western populations. Another possible explanation is the difference in the public knowledge regarding on the cardinal symptoms of migraine. Nevertheless, these speculations need to be validated, and further studies are needed on this issue with in various populations.

Nausea and vomiting are closely related symptoms and may have a sequential relationship; if nausea persists without improvement, it can progress to vomiting. However, nausea and vomiting have significant differences in terms of the severity of migraine. Nausea is the most common accompanying symptom of migraine and is observed in >90% of individuals with migraine [29]. Conversely, vomiting is less prevalent in these patients; compared to nausea, it is associated with more severe headaches [30]. Therefore, we considered nausea and vomiting separately to assess the MBS.

The present study found no significant association between migraine with aura and photophobia as the MBS. This is contradictory to the findings of a significant association between the two in the aforementioned Taiwanese study [7]. One possible reason for the discrepancy is the difference in the study settings. Our study evaluated the MBS in participants with migraine in a population-based sample, whereas the Taiwanese study assessed the MBS among patients with migraine who visited the hospital. The photophobia symptom is known to be more pronounced in migraine with aura than in MO. In a hospital-based study, patients with migraine with aura are more likely to select photophobia as the MBS, because their photophobia is more severe; however, participants from a population-based sample may be less likely to select photophobia, because their photophobia may be relatively mild.

This study first evaluated the MBS in participants with PM and demonstrated that its distribution in this population was different from that in participants with migraine, despite the order of the MBS frequency in PM being similar to that in migraine (Table 3). Furthermore, we observed no significant association between the MBS and the clinico-demographic characteristics of participants with PM. The discrepancy in the MBS between migraine and PM may be attributed to differences in the symptom severity and disability, which were milder in PM group than in the migraine group in this study (Table 1).

Investigation of the MBS in individuals with migraine and those with PM can help identify and manage their distress appropriately. Nausea was the most common MBS, followed by

phonophobia, in both migraine and PM. Active management through anti-emetics or the parenteral route can assist patients with nausea as the MBS. For individuals with phonophobia as the MBS, noise shielding (such as the use of ear plugs) can be beneficial.

This study has some limitations. First, the total participation rate was low. Nevertheless, we obtained a study sample that was proportional to the population distribution in Korea using a two-stage clustered random sampling method. The sociodemographic distribution did not differ significantly between the survey participants and the general Korean population. Second, we used the mTOQ-6 to evaluate the acute treatment response. However, we could not ensure that all participants received optimal treatment. The majority of guidelines recommend triptans as the first-line acute treatment. Nevertheless, its use in participants with migraine and those with PM was low. Notably, only 0.6% of the participants with migraine used triptans; this was lower than that reported in other population-based studies [31, 32]. Furthermore some participants did not consume any acute medications. Third, we used questionnaires to evaluate clinical information (such as medication usage, headache days, and severe headache days) through a web-based survey. This information can be assessed using more accurate or objective methods, such as diaries. However, it was difficult to adopt this approach in our study because of the large number of participants; thus, we investigated these data using a web-based survey. Fourth, we diagnosed migraine and PM using diagnostic modules. Although our diagnostic modules have a high sensitivity and specificity, some participants with migraine and those with PM could have been misclassified; consequently, this could have affected our results. However, given the distribution of the MBS in the participants with migraine and those with PM in this study, it is highly unlikely that nausea is not the most common MBS in both conditions.

The strengths of this study are the use of data from a large sample size proportional to the population distribution, evaluation of the headache type and response to acute treatment using validated instruments, and use of the MBS based on the diagnostic criteria for migraine in ICHD-3.

## Conclusions

Nausea was the predominant MBS, followed by photophobia and vomiting, in both migraine and PM. Phonophobia as the MBS was more common in migraine than in PM. However, nausea as the MBS was more common in PM than in migraine. Certain clinical features were significantly associated with the selection of the MBS for migraine. In contrast, there were no significant associations between the MBS and the clinical features in participants with PM. The response to acute treatment was not significantly associated with the MBS in both migraine and PM.

## Supporting information

**S1 Table. Sociodemographic distribution of the total Korean population, survey participants, and cases of migraine and PM.**
(DOCX)

**S2 Table. Distribution of associated symptoms in participants with migraine and those with PM.**
(DOCX)

**S3 Table. Use of acute medication classes in the migraine and PM groups.**
(DOCX)

## Author Contributions

**Conceptualization:** Seung Jae Kim, Min Kyung Chu.

**Data curation:** Seung Jae Kim.

**Formal analysis:** Kyung Min Kim.

**Investigation:** Min Kyung Chu.

**Methodology:** Kyung Min Kim.

**Project administration:** Min Kyung Chu.

**Resources:** Sue Hyun Lee, Soomi Cho.

**Supervision:** Min Kyung Chu.

**Validation:** Min Kyung Chu.

**Writing – original draft:** Seung Jae Kim, Hye Jeong Lee.

**Writing – review & editing:** Hye Jeong Lee, Min Kyung Chu.

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
