## [Decision Letter · Decision Letter 0]

14 Jun 2023

PONE-D-23-13392Most bothersome symptom in migraine and probable migraine: a population-based studyPLOS ONE

Dear Dr. Chu,

Thank you for submitting your manuscript to PLOS ONE. After careful consideration, we feel that it has merit but does not fully meet PLOS ONE’s publication criteria as it currently stands. Therefore, we invite you to submit a revised version of the manuscript that addresses the points raised during the review process.

We look forward to receiving your revised manuscript.

Kind regards,

Keisuke Suzuki, MD, PhD

Academic Editor

PLOS ONE

Journal Requirements:

   "This research was supported by a grant from the Korea Health Technology R&D Project through the Korea Health Industry Development Institute (KHIDI), funded by the Ministry of Health & Welfare, Republic of Korea (Grant No.: HV22C0106) and a National Research Foundation of Korea (NRF) grant from the Korean government (MSIT) (2022R1A2C1091767)."

   "M.K.C. was a site investigator for a multicenter trial sponsored by Biohaven Pharmaceuticals, Allergan Korea, and Ildong Pharmaceutical Company. He received lecture honoraria from Eli Lilly and Company, Handok-Teva, and Ildong Pharmaceutical Company over the past 24 months. He received grants from Yonsei University College of Medicine (6-2021-0229), the Korea Health Industry Development Institute (KHIDI) (Grant No.: HV22C0106), and a National Research Foundation of Korea (NRF) grant from the Korean government (MSIT) (2022R1A2C1091767). No other authors possess competing interests."

Additional Editor Comments:

Here are some interesting review comments from the peer reviewers.

Please revise your manuscript according to the reviewers' comments.

Reviewers' comments:

Reviewer's Responses to Questions

**Comments to the Author**

1. Is the manuscript technically sound, and do the data support the conclusions?

Reviewer #1: Yes

Reviewer #2: Yes

2. Has the statistical analysis been performed appropriately and rigorously? 

Reviewer #1: Yes

Reviewer #2: Yes

3. Have the authors made all data underlying the findings in their manuscript fully available?

Reviewer #1: Yes

Reviewer #2: Yes

4. Is the manuscript presented in an intelligible fashion and written in standard English?

Reviewer #1: Yes

Reviewer #2: Yes

5. Review Comments to the Author

Reviewer #1: Kim SJ et al. investigated what the common bothersome symptom (MBS) of Korean migraineurs are. Nausea was most common followed by phonophobia, vomiting and photophobia. I found the manuscript very well written with rigid data and appropriate citations providing another important information about Asians migraineurs which are different from Western population.

I only have few minor comments.

1. Association of BMI and photophobia reported by Winter AC et al. was interesting to know. Could you explain the pathophysiology (if not well known, your speculation) why high BMI may lead to higher percentage of photophobia?

2. Authors mentioned the response to acute treatment and its association with MBS. It is important for readers to know that the use of triptan was very low in the current cohort (noted in Supplementary table). I suggest mentioning the percentage of triptan users (around 0.3%?) in the main text.

3. I thought alternative way of showing % of Table 3, Table 4 and Sup Table 1 would be using number of patients in each category shown on the first row for denominators. I am not 100% sure which one would be more appropriate. You can leave it as current form if you think the current way is better, but please consider.

4. Sup Table 2: I was amazed that all probable migraine patients had at least one of the associated symptoms. By ICHD-3, if a patient meets category A, B, C and E, and not D (associated symptoms), he/she can be diagnosed as probable migraine (if not meeting definite TTH diagnosis). Could you explain why there was no patient without associated symptoms in probable migraine group (i.e. were patients without associated symptoms excluded)?

Reviewer #2: Ever since the US FDA proposed "most bothersome symptoms" (MBS) as a co-primary endpoint, in addition to clinical trial results, several large-scaled research studies have been published from both Eastern and Western countries. The concept of this paper aims to explore the differences between migraine and probable migraine (PM), which indeed represents the current scientific gap. The research findings also provide conclusive evidence of variations in MBS between Eastern and Western populations. The following are some issues for the authors to address:

1. In the Methods Section, the subjects' headache diagnoses were derived from the migraine diagnostic module, without further confirmation from headache specialists. As mentioned in the text, the migraine diagnostic module is not entirely accurate in diagnosing migraine and PM. If there are misdiagnoses, the association between MBS and headache could be affected. How does this impact the conclusions of the study?

2. In the Methods section, nausea and vomiting were treated as two distinct entities. However, according to ICHD-3, the criteria for recognizing migraine-associated symptoms are "at least one of the following: nausea and/or vomiting, photophobia and phonophobia." Therefore, nausea and vomiting can actually be considered as one item in the ICHD-3. In fact, these two symptoms often have a sequential relationship in many patients: if nausea persists without improvement, it can progress to vomiting. The authors are suggested to explain why they separated these two symptoms during analysis.

3. In the Results section, no association between visual aura and photophobia as an MBS, as shown in Tables 3 and 4, regardless of whether it was migraine or PM. However, Taiwanese study mentioned a higher frequency of photophobia as MBS in migraine with aura compared to migraine without aura and chronic migraine. As both studies focus on Asian populations, why is there such a difference?

4. In the Discussion section, it is mentioned that one possible explanation for the differences in MBS between Eastern and Western populations could be the varying prevalence of photophobia. However, differences in the prevalence of photophobia do not necessarily indicate the level of bothersomeness. If you hypothesize that "a symptom with lower prevalence is less likely to be chosen as MBS," it contradicts the findings of your study, where the majority of participants with migraine still selected nausea as MBS rather than phonophobia or photophobia, despite the proportions of these symptoms in the migraine population being ranked as phonophobia (80.6%) > photophobia (73.5%) > nausea (70.6%). This discrepancy may challenge your hypothesis.

5. In the Discussion section, you mentioned that your results demonstrated the suitability of MBS as a co-primary endpoint. However, the treatment response in this study was assessed using mTOQ-6, which solely focuses on headaches and not MBS [Headache. 2015 Apr;55(4):502-18.]. Consequently, using mTOQ-6 as the sole indicator for treatment response in this study does not allow for a determination of whether MBS is suitable or not as a co-primary endpoint. Hence, the concluding statement in the abstract and the 7th paragraph of the discussion may be an overinterpretation of the results.

6. PLOS authors have the option to publish the peer review history of their article (what does this mean?). If published, this will include your full peer review and any attached files.

Reviewer #1: No

Reviewer #2: No

---

## [Author Response · Author response to Decision Letter 0]

16 Jul 2023

▨ Reviewer #1: Kim SJ et al. investigated what the common bothersome symptom (MBS) of Korean migraineurs are. Nausea was most common followed by phonophobia, vomiting and photophobia. I found the manuscript very well written with rigid data and appropriate citations providing another important information about Asians migraineurs which are different from Western population.

I only have few minor comments.

1. Association of BMI and photophobia reported by Winter AC et al. was interesting to know. Could you explain the pathophysiology (if not well known, your speculation) why high BMI may lead to higher percentage of photophobia?

☞ Response 

Thank you for your comment. Currently, photophobia in migraine is believed to be attributed to the interactions among the visual, trigeminal, and autonomic systems [1, 2]. The activity of the trigeminal system is increased in individuals with obesity; therefore, the increase in photophobia in these individuals may be attributed to an increased input from the trigeminal system. Autonomic dysfunction is also noted in obesity and can affect the photophobia symptoms [3].

Nevertheless, the description of the association between photophobia and the body mass index has been removed from the revised manuscript; this is because it is unlikely to account for the differences in the most bothersome symptom (MBS) between the Asian and Western studies (see page 25, lines 2–7). 

2. Authors mentioned the response to acute treatment and its association with MBS. It is important for readers to know that the use of triptan was very low in the current cohort (noted in Supplementary table). I suggest mentioning the percentage of triptan users (around 0.3%?) in the main text.

☞ Response 

Thank you for your valuable insight. In accordance with your comment, we have included the following text in the Discussion section (page 27, lines 14–16): “Notably, only 0.6 % of the participants with migraine used triptans; this was lower than that reported in other population-based studies [4, 5].”

3. I thought alternative way of showing % of Table 3, Table 4 and Sup Table 1 would be using number of patients in each category shown on the first row for denominators. I am not 100% sure which one would be more appropriate. You can leave it as current form if you think the current way is better, but please consider.

☞ Response 

We sincerely thank you for your valuable suggestion regarding data presentation in Table 3, Table 4, and S1 Table. We appreciate this insight; however, we did not add the denominators as suggested, because doing so would lead to redundancy and make the text more complicated. Instead, we have added the following footnote to these tables for improved clarity: “Note: Data are expressed as n (%) or median (25 %–75 %)”

4. Sup Table 2: I was amazed that all probable migraine patients had at least one of the associated symptoms. By ICHD-3, if a patient meets category A, B, C and E, and not D (associated symptoms), he/she can be diagnosed as probable migraine (if not meeting definite TTH diagnosis). Could you explain why there was no patient without associated symptoms in probable migraine group (i.e. were patients without associated symptoms excluded)?

☞ Response 

Thank you for your comment. According to the International Classification of Headache Disorders, Third Edition (ICHD-3), probable migraine (PM) is diagnosed in patients who do not fulfill all but one criterion for migraine diagnosis. Among the 339 participants with PM in our study, 337 and 2 were diagnosed with PM based on missing typical duration and missing typical accompanying symptoms, respectively. Therefore, almost all participants with PM could report the MBS in our study. Accordingly, we have included the following description on the diagnostic profiles of PM as follows (page 11, lines 6–8): “Among those with PM, 337 and 2 were diagnosed based on missing typical duration and missing typical accompanying symptoms, respectively.” 

Regarding the diagnostic profile of PM, studies have revealed that a missing typical duration was the most common cause of PM diagnosis, while missing typical accompanying symptoms were the least common causes of PM diagnosis [6, 7]. 

[REFERENCES]

1. Noseda R, Copenhagen D, Burstein R. Current understanding of photophobia, visual networks and headaches. Cephalalgia. 2019;39(13):1623-34. Epub 20180625. doi: 10.1177/0333102418784750. PubMed PMID: 29940781; PubMed Central PMCID: PMCPMC6461529.

2. Rossi HL, Luu AK, DeVilbiss JL, Recober A. Obesity increases nociceptive activation of the trigeminal system. Eur J Pain. 2013;17(5):649-53. Epub 20121016. doi: 10.1002/j.1532-2149.2012.00230.x. PubMed PMID: 23070979; PubMed Central PMCID: PMCPMC4275045.

3. Guarino D, Nannipieri M, Iervasi G, Taddei S, Bruno RM. The Role of the Autonomic Nervous System in the Pathophysiology of Obesity. Front Physiol. 2017;8:665. Epub 20170914. doi: 10.3389/fphys.2017.00665. PubMed PMID: 28966594; PubMed Central PMCID: PMCPMC5606212.

4. Chu MK, Buse DC, Bigal ME, Serrano D, Lipton RB. Factors associated with triptan use in episodic migraine: results from the American Migraine Prevalence and Prevention Study. Headache. 2012;52(2):213-23. doi: 10.1111/j.1526-4610.2011.02032.x. PubMed PMID: 22413150.

5. Hirata K, Ueda K, Komori M, Zagar AJ, Selzler KJ, Nelson AM, et al. Comprehensive population-based survey of migraine in Japan: results of the ObserVational Survey of the Epidemiology, tReatment, and Care Of MigrainE (OVERCOME [Japan]) study. Curr Med Res Opin. 2021;37(11):1945-55. Epub 20210922. doi: 10.1080/03007995.2021.1971179. PubMed PMID: 34429000.

6. Kim BK, Chung YK, Kim JM, Lee KS, Chu MK. Prevalence, clinical characteristics and disability of migraine and probable migraine: a nationwide population-based survey in Korea. Cephalalgia. 2013;33(13):1106-16. Epub 20130424. doi: 10.1177/0333102413484990. PubMed PMID: 23615490.

7. Lantéri-Minet M, Valade D, Géraud G, Chautard MH, Lucas C. Migraine and probable migraine--results of FRAMIG 3, a French nationwide survey carried out according to the 2004 IHS classification. Cephalalgia. 2005;25(12):1146-58. doi: 10.1111/j.1468-2982.2005.00977.x. PubMed PMID: 16305603.

▨ Reviewer #2: Ever since the US FDA proposed "most bothersome symptoms" (MBS) as a co-primary endpoint, in addition to clinical trial results, several large-scaled research studies have been published from both Eastern and Western countries. The concept of this paper aims to explore the differences between migraine and probable migraine (PM), which indeed represents the current scientific gap. The research findings also provide conclusive evidence of variations in MBS between Eastern and Western populations. The following are some issues for the authors to address:

1. In the Methods Section, the subjects' headache diagnoses were derived from the migraine diagnostic module, without further confirmation from headache specialists. As mentioned in the text, the migraine diagnostic module is not entirely accurate in diagnosing migraine and PM. If there are misdiagnoses, the association between MBS and headache could be affected. How does this impact the conclusions of the study?

☞ Response 

Thank you for your valuable insight. We agree with you that the participants in our study could have been misdiagnosed, and this could have affected our results. However, our diagnostic modules for migraine and probable migraine (PM) had high accuracies of 93.8 % and 91.0 %, respectively [8]. Based on the distribution of the most bothersome symptom (MBS) in participants with migraine and those with PM in this study (Tables 2 and 3), it is highly unlikely that nausea was not the most common MBS. We have addressed this issue as a limitation of this study in the Discussion section as follows (page 27, lines 22–24 through page 28, lines 1–2): “Fourth, we diagnosed migraine and PM using diagnostic modules. Although our diagnostic modules have a high sensitivity and specificity, some participants with migraine and those with PM could have been misclassified; consequently, this could have affected our results. However, given the distribution of the MBS in the participants with migraine and those with PM in this study, it is highly unlikely that nausea is not the most common MBS in both conditions.” 

2. In the Methods section, nausea and vomiting were treated as two distinct entities. However, according to ICHD-3, the criteria for recognizing migraine-associated symptoms are "at least one of the following: nausea and/or vomiting, photophobia and phonophobia." Therefore, nausea and vomiting can actually be considered as one item in the ICHD-3. In fact, these two symptoms often have a sequential relationship in many patients: if nausea persists without improvement, it can progress to vomiting. The authors are suggested to explain why they separated these two symptoms during analysis.

☞ Response 

Thank you very much for your important comment. We agree that nausea and vomiting are closely related symptoms. However, both have different significance regarding the severity of migraine. Nausea is the most common accompanying symptom of migraine and presents in >90% of the affected individuals [9]. In contrast, vomiting is less prevalent in these individuals; compared to nausea, it is associated with more severe headache pain [10]. Therefore, we considered nausea and vomiting as separate symptoms to assess the MBS. We have clarified this in the Discussion section as follows (page 25, lines 19–24 through page 26, line 1): “Nausea and vomiting are closely related symptoms and may have a sequential relationship; if nausea persists without improvement, it can be progress to vomiting. However, nausea and vomiting have significant differences in terms of the severity of migraine. Nausea is the most common accompanying symptom of migraine and is observed in >90 % of the individuals with migraine [9]. Conversely, vomiting is less prevalent in these patients; compared to nausea, it is associated with more severe headaches [10]. Therefore, we considered nausea and vomiting separately to assess the MBS.” 

3. In the Results section, no association between visual aura and photophobia as an MBS, as shown in Tables 3 and 4, regardless of whether it was migraine or PM. However, Taiwanese study mentioned a higher frequency of photophobia as MBS in migraine with aura compared to migraine without aura and chronic migraine. As both studies focus on Asian populations, why is there such a difference?

☞ Response 

Thank you for raising this important query. The flagged discrepancy between our findings and the findings of the previous Taiwanese study could be attributed to the differences in the study settings between the two. We have addressed this in the Discussion section as follows (pages 26, lines 2–11): “The present study found no significant association between migraine with aura and photophobia as the MBS. This is contradictory to the findings of a significant association between the two in the aforementioned Taiwanese study [11]. One possible reason for the discrepancy is the difference in the study settings. Our study evaluated the MBS in participants with migraine in a population-based sample, whereas the Taiwanese study assessed the MBS among patients with migraine who visited the hospital. The photophobia symptom is known to be more pronounced in migraine with aura than in MO. In a hospital-based study, patients with migraine with aura are more likely to select photophobia as the MBS, because their photophobia is more severe; however, participants from a population-based sample may be less likely to select photophobia, because their photophobia may be relatively mild.” 

4. In the Discussion section, it is mentioned that one possible explanation for the differences in MBS between Eastern and Western populations could be the varying prevalence of photophobia. However, differences in the prevalence of photophobia do not necessarily indicate the level of bothersomeness. If you hypothesize that "a symptom with lower prevalence is less likely to be chosen as MBS," it contradicts the findings of your study, where the majority of participants with migraine still selected nausea as MBS rather than phonophobia or photophobia, despite the proportions of these symptoms in the migraine population being ranked as phonophobia (80.6%) > photophobia (73.5%) > nausea (70.6%). This discrepancy may challenge your hypothesis.

☞Response 

Thank you for your comment. You have rightly noted that “a symptom with lower prevalence is less likely to be chosen as MBS” is not an adequate explanation for the regional differences in the MBS between the Asian and Western countries. Accordingly, we have revised this portion in the Discussion section as follows (pages 25, lines 2–7): “The reasons underlying these regional differences in the MBS are difficult to elucidate. One possible reason is the difference in the tolerance thresholds for each migraine symptom between the Asian and Western populations. Another possible explanation is the difference in the public knowledge on the cardinal symptoms of migraine. Nevertheless, these speculations need to be validated, and further studies are needed in various populations.” 

5. In the Discussion section, you mentioned that your results demonstrated the suitability of MBS as a co-primary endpoint. However, the treatment response in this study was assessed using mTOQ-6, which solely focuses on headaches and not MBS [Headache. 2015 Apr;55(4):502-18.]. Consequently, using mTOQ-6 as the sole indicator for treatment response in this study does not allow for a determination of whether MBS is suitable or not as a co-primary endpoint. Hence, the concluding statement in the abstract and the 7th paragraph of the discussion may be an overinterpretation of the results.

☞Response

Thank you very much for flagging this with us. We evaluated the acute treatment response using the migraine Treatment Optimization Questionnaire-6 (mTOQ-6) in the present study. mTOQ-6 was developed based on items regarding treatment needs, disability, and quality of life in patients with migraine as well as the patient preference and satisfaction [12]. The utility of mTOQ-6 was validated by comparing 2-hour pain freedom and 24-hour pain relief [13, 14]. Nevertheless, it was not validated by comparing the changes in the MBS. Therefore, we agree with you that our concluding statements could represent an overinterpretation. Accordingly, we have removed these portions from the Abstract (page 3, lines 20–21) and the Discussion section (pages 25, lines 7–18).

[REFERENCES]

8. Kim KM, Kim AR, Lee W, Jang BH, Heo K, Chu MK. Development and validation of a web-based headache diagnosis questionnaire. Sci Rep. 2022;12(1):7032. Epub 2022/04/30. doi: 10.1038/s41598-022-11008-y. PubMed PMID: 35488015; PubMed Central PMCID: PMCPMC9052186 Novartis International AG, and Eli Lilly and Company. He worked as an advisory member for Teva and has received lecture honoraria from Allergan Korea, Handok-Teva, and Yuyu Pharmaceutical Company in the past 24 months. He received grants from Yonsei University College of Medicine (2018-32-0037) and National Research Foundation of Korea (2019R1F1A1053841). The other authors declare no conflicts of interest.

9. Silberstein SD. Migraine symptoms: results of a survey of self-reported migraineurs. Headache. 1995;35(7):387-96. doi: 10.1111/j.1526-4610.1995.hed3507387.x. PubMed PMID: 7672955.

10. Kelman L, Tanis D. The relationship between migraine pain and other associated symptoms. Cephalalgia. 2006;26(5):548-53. doi: 10.1111/j.1468-2982.2006.01075.x. PubMed PMID: 16674763.

11. Tu YH, Wang YF, Yuan H, Chen SP, Tzeng YS, Chen WT, et al. Most bothersome symptoms in patients with migraine: A hospital-based study in Taiwan. Headache. 2022;62(5):596-603. Epub 2022/04/26. doi: 10.1111/head.14308. PubMed PMID: 35467015.

12. Lipton RB, Kolodner K, Bigal ME, Valade D, Láinez MJ, Pascual J, et al. Validity and reliability of the Migraine-Treatment Optimization Questionnaire. Cephalalgia. 2009;29(7):751-9. Epub 20090223. doi: 10.1111/j.1468-2982.2008.01786.x. PubMed PMID: 19239676.

13. Ezzati A, Fanning KM, Reed ML, Lipton RB. Predictors of treatment-response to caffeine combination products, acetaminophen, acetylsalicylic acid (aspirin), and nonsteroidal anti-inflammatory drugs in acute treatment of episodic migraine. Headache. 2023;63(3):342-52. Epub 20230207. doi: 10.1111/head.14459. PubMed PMID: 36748728.

14. Ezzati A, Buse DC, Fanning KM, Reed ML, Martin VT, Lipton RB. Predictors of treatment-response to acute prescription medications in migraine: Results from the American Migraine Prevalence and Prevention (AMPP) Study. Clin Neurol Neurosurg. 2022;223:107511. Epub 20221103. doi: 10.1016/j.clineuro.2022.107511. PubMed PMID: 36395587.

---

## [Decision Letter · Decision Letter 1]

26 Jul 2023

Most bothersome symptom in migraine and probable migraine: a population-based study

PONE-D-23-13392R1

Dear Dr. Chu,

We’re pleased to inform you that your manuscript has been judged scientifically suitable for publication and will be formally accepted for publication once it meets all outstanding technical requirements.

Kind regards,

Keisuke Suzuki, MD, PhD

Academic Editor

PLOS ONE

Additional Editor Comments (optional):

Reviewers' comments:

Reviewer's Responses to Questions

**Comments to the Author**

1. If the authors have adequately addressed your comments raised in a previous round of review and you feel that this manuscript is now acceptable for publication, you may indicate that here to bypass the “Comments to the Author” section, enter your conflict of interest statement in the “Confidential to Editor” section, and submit your "Accept" recommendation.

Reviewer #1: All comments have been addressed

Reviewer #2: All comments have been addressed

2. Is the manuscript technically sound, and do the data support the conclusions?

Reviewer #1: Yes

Reviewer #2: Yes

3. Has the statistical analysis been performed appropriately and rigorously? 

Reviewer #1: I Don't Know

Reviewer #2: Yes

4. Have the authors made all data underlying the findings in their manuscript fully available?

Reviewer #1: Yes

Reviewer #2: Yes

5. Is the manuscript presented in an intelligible fashion and written in standard English?

Reviewer #1: Yes

Reviewer #2: Yes

6. Review Comments to the Author

Reviewer #1: I have no further comments on this manuscript, and I think it is acceptable. Thank you very much for your hard work.

Reviewer #2: The authors have addressed my comments adequately in the revised manuscipt. I do not have further comments.

7. PLOS authors have the option to publish the peer review history of their article (what does this mean?). If published, this will include your full peer review and any attached files.

Reviewer #1: No

Reviewer #2: No

---

## [Editor Report · Acceptance letter]

28 Jul 2023

PONE-D-23-13392R1 

Most bothersome symptom in migraine and probable migraine: A population-based study 

Dear Dr. Chu:

I'm pleased to inform you that your manuscript has been deemed suitable for publication in PLOS ONE. Congratulations! Your manuscript is now with our production department. 

Kind regards, 

on behalf of

Dr. Keisuke Suzuki 

Academic Editor

PLOS ONE